# Do frailty measures improve prediction of mortality and morbidity following transcatheter aortic valve implantation? An analysis of the UK TAVI registry

Glen P Martin,[1] Matthew Sperrin,[1] Peter F Ludman,[2] Mark A deBelder,[3] Mark Gunning,[4,5] John Townend,[2] Simon R Redwood,[6] Umesh T Kadam,[4,5] Iain Buchan,[1,7] Mamas A Mamas[1,4,5]

For numbered affiliations see end of article.

**Correspondence to**
Dr Glen P Martin;
glen.martin@manchester.ac.uk

## ABSTRACT

**Objectives** Previous studies indicate frailty to be associated with poor outcomes following transcatheter aortic valve implantation (TAVI), but there is limited evidence from multicentre registries. The aim was to investigate the independent association of frailty with TAVI outcomes, and the prognostic utility of adding frailty into existing clinical prediction models (CPMs).

**Design** The UK TAVI registry incorporated three frailty measures since 2013: Canadian Study of Health and Ageing, KATZ and poor mobility. We investigated the associations between these frailty measures with short-term and long-term outcomes, using logistic regression to estimate multivariable adjusted ORs, and Cox proportional hazards models to explore long-term survival. We compared the predictive performance of existing TAVI CPMs before and after updating them to include each frailty measure.

**Setting** All patients who underwent a TAVI procedure in England or Wales between 2013 and 2014.

**Participants** 2624 TAVI procedures were analysed in this study.

**Primary and secondary outcomes** The primary endpoints in this study were 30-day mortality and long-term survival. The Valve Academic Research Consortium (VARC)-2 composite early safety endpoint was considered as a secondary outcome.

**Results** KATZ <6 (OR 2.10, 95% CI 1.39 to 3.15) and poor mobility (OR 2.15, 95% CI 1.41 to 3.28) predicted 30-day mortality after multivariable adjustment. All frailty measures were associated with increased odds of the VARC-2 composite early safety endpoint. We observed a significant increase in the area under the receiver operating characteristic curves by approximately 5% after adding KATZ <6 or poor mobility into the TAVI CPMs. Risk stratification agreement was significantly improved by the addition of each frailty measure, with an increase in intraclass correlation coefficient of between 0.15 and 0.31.

**Conclusion** Frailty was associated with worse outcomes following TAVI, and incorporating frailty metrics significantly improved the predictive performance of existing CPMs. Physician-estimated frailty measures could aid TAVI risk stratification, until more objective scales are routinely collected.

## Strengths and limitations of this study

► An analysis of the association between three frailty measures and clinical outcomes following transcatheter aortic valve implantation (TAVI).
► The study used data from the UK TAVI registry, including all consecutive patients who underwent TAVI in UK between 2013 and 2014.
► A large, contemporary study, which investigated the impact of frailty on outcomes and mortality prediction post-TAVI from a national perspective.
► Only three subjective frailty measures were available, with objective frailty tests not recorded.
► This retrospective analysis could not compare outcomes across frailty groups in patients with untreated aortic stenosis, or those undergoing surgical replacement.

## INTRODUCTION

Transcatheter aortic valve implantation (TAVI) has emerged as an effective treatment strategy for patients with aortic stenosis who are intermediate-to-high operative risk.[1–3] To this end, surgical clinical prediction models (CPMs), such as the EuroSCORE or the Society of Thoracic Surgeons (STS) model, have previously been used to estimate risk, but TAVI-specific CPMs are emerging from national registries: examples include the FRANCE-2 model,[4] the OBSERVANT model[5] and the American College of Cardiology (ACC) Transcatheter Valve Therapy model.[6] However, these models do not usually incorporate measures of frailty and disability into geriatric prescreening risk assessment.

Frailty is an emerging concept in clinical science and is defined as an age-related decline in the resilience to stressors caused by deterioration in multiple physiological systems.[7] Consequently, there is a growing evidence base of associations between frailty and poor outcomes after TAVI.[8–13] Indeed, patient's frailty, dependencies in activities of daily living (ADL) and cognitive function are increasingly being considered in the decision-making process for TAVI eligibility.[14] However, measures of frailty are rarely incorporated into many of the currently available CPMs, which has potentially contributed to the moderate predictive performance of such models outside their development datasets.[15–17] The recent recording of measures relating to frailty/disability in national cohorts presents novel opportunities to incorporate them into future iterations of existing TAVI CPMs.

Therefore, the aim was to investigate the effect of frailty and disability on mortality and morbidity in a national TAVI cohort and to examine the prognostic utility of adding frailty/disability into existing TAVI CPMs.

## METHODS
### UK TAVI registry
The UK TAVI registry prospectively collects 95 variables (including patient demographics, risk factors for intervention and within-hospital adverse outcomes), for every TAVI procedure conducted in the UK.[18] This analysis used data from patients in England and Wales, for whom all-cause mortality was linked from the Office for National Statistics. The study period for this analysis was January 2013 to December 2014, corresponding to the time in which the UK registry recorded three variables that related to frailty/disability. Specifically, the registry recorded the Canadian Study of Health and Ageing (CSHA)-estimated frailty scale,[19] KATZ ADL dependency[20] and a physician-estimated poor mobility (as defined in the EuroSCORE II model[21]). Frailty groups were defined separately across the three measures with the full definitions of each group given in table 1. Dichotomising CSHA and KATZ scores into two categories was based on the empirical median value (CSHA='apparently vulnerable' and KATZ=6), and on the original publication of CSHA.[19] Additionally, for this analysis, we defined a composite score to incorporate information from all three of the measures, with the definition given in table 1.

### Study endpoints
The primary endpoints in this study were 30-day mortality and long-term survival. As a secondary endpoint, we investigated the Valve Academic Research Consortium (VARC)-2 composite early safety endpoint, which is defined as a failure in any of the following outcomes by 30 days: all-cause mortality, stroke, life-threatening bleeding, stage 2/3 acute kidney injury, coronary artery obstruction requiring intervention, major vascular complications or valve-related dysfunction requiring a repeat procedure.[14]

| Table 1 | Frailty grouping definitions by each frailty/disability measure | |
|---|---|---|
| **Frailty/disability measure** | **Scoring system** | **Frailty definition used in this analysis** |
| CSHA* | A physician-estimated frailty score, based on the following options: (1) very fit, (2) well,; (3) well with treated comorbid disease, (4) apparently vulnerable, (5) mildly frail with limited dependence for activities of daily living (ADL), (6) moderately frail requiring help with ADL, (7) severely frail being completely dependent. | CSHA options of 5–7 were used to define CSHA frail patients; options 1–4 were classed as CSHA non-frail. |
| KATZ* | 0–6 points scale assessing dependency in the following ADL: (1) bathing, (2) dressing, (3) toileting, (4) transferring, (5) continence, (6) Feeding. | Any patient with a KATZ score <6 points compared with those with KATZ=6. |
| Poor mobility | A physician-estimated indication of any severe impairment of mobility that is secondary to musculoskeletal or neurological dysfunction. | Any patient defined as having poor mobility compared with those defined as having normal mobility. |
| Composite score | Combination of CSHA, KATZ and poor mobility. | Defined as the following: (1) Non-frail if a patient is estimated as not frail by CSHA, KATZ and poor mobility. (2) Moderately frail if a patient is estimated as frail by at least one (but not all) considered frailty measures. (3) Severely frail if a patient is classed as frail across all three of CSHA, KATZ and poor mobility. |

*Dichotomising of CSHA and KATZ into two groups was based on the median level observed in the TAVI registry (see online supplementary figures 1 and 2 for the distribution of these scores).
CSHA, Canadian Study of Health and Ageing; TAVI, transcatheter aortic valve implantation.

## Statistical analysis

Every variable with missing data was imputed using multiple imputation that generated 10 imputed datasets.[22] The imputation model for each variable included the majority of other variables in the UK TAVI registry and all considered endpoints.[23] Imputed frailty measures and outcomes were not used, and instead were returned to the original (missing) values.[24] Subsequently, patients missing data on CSHA, KATZ, poor mobility and/or missing life status were removed from the analysis; patients with missing VARC-2 composite early safety were only removed from the analysis of that outcome. All analyses were undertaken in each imputed dataset separately, with the results pooled according to Rubin's rules.[22]

For exploratory analysis, we obtained spatial maps of England at a National Health Service (NHS) regional level (a high-level geographical structure partitioning England into 13 regions) and of Wales, from the Office for National Statistics. By linking the clinical commission group of each patient to the corresponding NHS region, we calculated the proportion of frail patients in the study population across England and Wales.

Odds Ratios (ORs) for binary endpoints were estimated using logistic regression, with both unadjusted and adjusted ORs reported. Multivariable models adjusted for the following variables: total centre volume, age, sex, diabetes status, smoking status, creatinine, renal failure, previous myocardial infarction, pulmonary disease, neurological disease, extracardiac arteriopathy, calcification of ascending aorta, atrial fibrillation/flutter, previous cardiac surgery, previous percutaneous coronary intervention, height, weight, critical preoperative status, Canadian Cardiovascular Society grading class 4, New York Heart Association (NYHA) class ≥III, aortic valve area and peak gradient, left ventricular ejection fraction (LVEF) <50%, one or more diseases vessels, left main stem disease, non-elective procedure indication and non-transfemoral access indication.

Long-term survival was assessed non-parametrically using Kaplan-Meier plots and strata were compared using the log-rank test. Multivariable adjustment was performed using Cox proportional hazards models, using the same variables as above to adjust the frailty indication hazard ratio (HR) for measured confounding. To account for non-proportional hazards, time was split into strata of 0–180 days, 180–365 days and >365 days, with frailty-indication-by-time-indicator interactions included in the Cox proportional hazards models.

The effect on predictive performance of adding CSHA-estimated frailty, KATZ ADL dependency or physician-estimated poor mobility into the existing TAVI CPMs (FRANCE-2, OBSERVANT and ACC) was considered using model updating techniques.[25–27] Mathematical details of model updating techniques are given in the supplementary methods. In short, CSHA, KATZ and poor mobility were added (both separately and sequentially for each measure) into an existing CPM by fitting a logistic regression model of 30-day mortality, with both the linear predictor of the considered TAVI CPM and at least one of the frailty measures as covariates. The likelihood ratio test (LRT) was used to test for significant improvement in model fit before and after model updating. Additionally, the area under the receiver operating characteristic curve (AUC) was calculated for each model and compared using a DeLong comparison.[28] Finally, intraclass correlation coefficient (ICC) estimates and their 95% CIs were calculated based on an absolute-agreement, two-way random effects model to indicate the risk-prediction agreement between the TAVI CPMs both before and after the addition of CSHA, KATZ or poor mobility.

R V.3.4.0[29] was used for all statistical analyses. Graphical plots were made using the ggplot2 package[30] and the mice package was used for the multiple imputation.[31]

## Patient and public involvement

No patients were involved in the design or conducting of this study.

## RESULTS

Between January 2013 and December 2014, n=3073 patients underwent a TAVI procedure in England and Wales. After excluding 104 patients with missing life status and 345 patients with any missing frailty measure, 2624 patients were analysed in this study. Due to the relative high proportion of exclusions (14.6%), we examined baseline characteristics of included patients compared with those excluded (online supplementary table 1). Most of the baseline characteristics were similar between the study group and the excluded group, with the exception of access route, proportion of calcified aorta and NYHA ≥III.

A total of 1043 patients (39.7%) had CSHA-estimated frailty (online supplementary figure 1), 846 patients (32.2%) had ADL dependency (KATZ <6) (online supplementary figure 2) and 591 patients (22.5%) had physician-estimated poor mobility. Figure 1 shows a Venn diagram of the overlap in classification across the three measures. The central segment of the Venn diagram (n=297) is the group defined as 'severely frail' by the composite score, with the other segments (n=1169) showing the 'moderately frail' group. The disagreement in figure 1 may relate to both the relative imprecision and subjectivity of each frailty measure and that the different measures assess disparate aspects of frailty.

Baseline characteristics and the proportion of missing data for the whole cohort are presented in table 2. Comparisons of baseline characteristics across CSHA-estimated frailty, KATZ ADL dependency and physician-estimated poor mobility groups are given in online supplementary tables 2–4, respectively. CSHA-estimated frail patients were more likely female (p<0.001), with higher proportions of NYHA ≥III (p<0.001), LVEF <50% (p=0.001) and non-elective procedures (p<0.001), but with lower numbers of CSHA-estimated frail patients having a previous cardiac surgery (p<0.001).

CSHA-estimated frail (n=1043)
KATZ<6 (n=846)
Physician-estimated poor mobility (n=591)

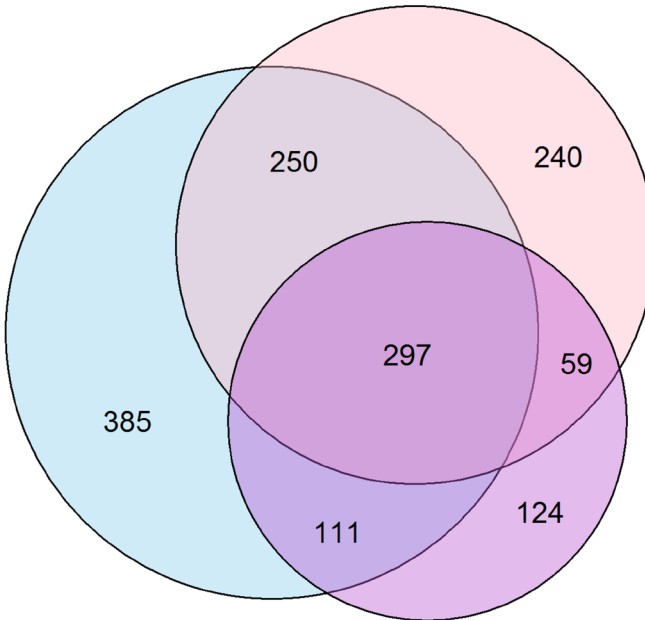

**Figure 1** Venn diagram showing the overlap in the different definitions of frailty/disability across CSHA-estimated frailty, KATZ <6 and physician-estimated poor mobility; the sizes of each segment are proportional to the segment sample sizes. CSHA, Canadian Study of Health and Ageing.

Similar differences were observed between KATZ ADL dependency (online supplementary table 3) and between physician-estimated poor mobility groups (online supplementary table 4).

There was large spatial heterogeneity in the proportion of TAVI procedures conducted in patients estimated as having poor mobility, KATZ <6 or CSHA-estimated frailty (online supplementary figure 3). For example, proportions of physician-estimated poor mobility patients across NHS regions ranged from 6.5% to 42.7%; NHS England North (Lancashire and Greater Manchester) had the highest proportion of patients with KATZ <6 and CSHA-estimated frailty, while NHS England South (Wessex) had the highest proportion of physician-estimated poor mobility patients.

### Frailty and post-TAVI outcomes

Incremental increases in components of CSHA and KATZ were associated with increased crude 30-day mortality rates (online supplementary figures 2 and 3). Specifically, CSHA-estimated frail patients had higher crude 30-day mortality, but this was not significant after multivariable adjustment (table 3). In contrast, physician-estimated poor mobility and KATZ <6 were significantly associated with increased odds of 30-day mortality after multivariable adjustment. All three of the considered frailty measures were independently associated with increased VARC-2 composite early safety endpoint (table 3). Similarly, those

**Table 2** Baseline characteristics and number of missing data within the whole cohort

| Variable | Whole cohort (n=2624) | Missing, n (%) |
|---|---|---|
| Age, mean (SD) | 81.2 (7.58) | 0 (0.00) |
| Female, n (%) | 1192 (45.4) | 1 (0.04) |
| Diabetic, n (%) | 641 (24.4) | 2 (0.08) |
| Smoker, n (%) | 1316 (50.2) | 45 (1.71) |
| Creatinine, mean (SD) | 110.9 (60.5) | 6 (0.23) |
| Renal failure*, n (%) | 143 (5.45) | 10 (0.38) |
| Previous MI, n (%) | 558 (21.3) | 2 (0.08) |
| Pulmonary disease, n (%) | 791 (30.1) | 3 (0.11) |
| Neurological disease, n (%) | 441 (16.8) | 1 (0.04) |
| Extracardiac arteriopathy, n (%) | 555 (21.2) | 5 (0.19) |
| Calcification of ascending aorta, % (n) | 378 (14.4) | 22 (0.84) |
| Atrial fibrillation, n (%) | 700 (26.7) | 23 (0.88) |
| Previous cardiac surgery, n (%) | 799 (30.4) | 5 (0.19) |
| Previous PCI, n (%) | 524 (20.0) | 2 (0.08) |
| Height, mean (SD) | 1.64 (0.10) | 19 (0.72) |
| Weight, mean (SD) | 75.0 (17.1) | 18 (0.69) |
| CCS class 4, n (%) | 24 (0.91) | 3 (0.11) |
| NYHA ≥III, n (%) | 1985 (75.6) | 6 (0.23) |
| Aortic valve area, mean (SD) | 0.69 (0.23) | 147 (5.60) |
| Aortic valve peak gradient, mean (SD) | 71.2 (26.2) | 129 (4.92) |
| LVEF <50%, n (%) | 948 (36.1) | 12 (0.46) |
| One or more diseased vessels, n (%) | 1058 (40.3) | 47 (1.79) |
| Left main stem disease, n (%) | 103 (3.93) | 75 (2.86) |
| Non-elective procedure, n (%) | 365 (13.9) | 2 (0.08) |
| Access site | | |
| Transfemoral, n (%) | 2127 (81.1) | 3 (0.11) |
| Transapical, n (%) | 249 (9.49) | 3 (0.11) |
| Subclavian, n (%) | 85 (3.24) | 3 (0.11) |
| Other, n (%) | 160 (6.10) | 3 (0.11) |

*Defined as creatinine >200 µmol/L or dialysis for renal failure.
CCS, Canadian Cardiovascular Society; LVEF, left ventricular ejection fraction; MI, myocardial Infarction; NYHA, New York Heart Association; PCI, percutaneous coronary intervention.

defined as frail across all three of CSHA, KATZ and poor mobility (ie, severely frail within the composite score) had significantly higher multivariable adjusted odds of 30-day mortality and composite early safety compared with patients defined as non-frail by the composite score (online supplementary table 5).

**Table 3** Short-term outcomes across CSHA-estimated frailty, KATZ ADL dependency and physician-estimated poor mobility groups

| Outcome | CSHA-estimated frail (n=1043) | CSHA-estimated non-frail (n=1581) | Univariable OR (95% CI)* | Multivariable OR (95% CI)* |
|---|---|---|---|---|
| 30-day mortality | 57/1043 (5.47%) | 51/1581 (3.23%) | **1.73 (1.18 to 2.55)** | 1.46 (0.96 to 2.23) |
| Early safety | 187/1014 (18.44%) | 190/1540 (12.34%) | **1.61 (1.29 to 2.00)** | **1.45 (1.14 to 1.84)** |
| Outcome | KATZ <6 (n=846) | KATZ=6 (n=1778) | Univariable OR (95% CI)* | Multivariable OR (95% CI)* |
| 30-day mortality | 58/846 (6.86%) | 50/1778 (2.81%) | **2.54 (1.73 to3.75)** | **2.10 (1.39 to3.15)** |
| Early safety | 150/827 (18.14%) | 227/1727 (13.14%) | **1.46 (1.17 to 1.83)** | **1.28 (1.01 to 1.63)** |
| Outcome | Poor mobility (n=591) | Normal mobility (n=2033) | Univariable OR (95% CI)* | Multivariable OR (95% CI)* |
| 30-day mortality | 46/591 (7.78%) | 62/2033 (3.05%) | **2.68 (1.81 to 3.98)** | **2.15 (1.41 to 3.28)** |
| Early safety | 116/571 (20.32%) | 261/1983 (13.16%) | **1.68 (1.32 to 2.14)** | **1.45 (1.12 to 1.88)** |

*Bold items indicate significant results.
ADL, activities of daily living; CSHA, Canadian Study of Health and Ageing.

Overall survival was 95.9%, 85.2% and 76.3% at 30 days, 1 year and 2 years, respectively. Survival was significantly worse for patients with CSHA-estimated frailty (p<0.0001), KATZ <6 (p<0.0001), physician-estimated poor mobility (p<0.001), and across the composite frailty groups (p<0.0001) (figure 2). The multivariable time-dependent Cox proportional hazards models indicated that hazards of mortality were significantly higher for those with KATZ <6 or physician-estimated poor mobility within 1-year post-TAVI (table 4); after 1 year, multivariable

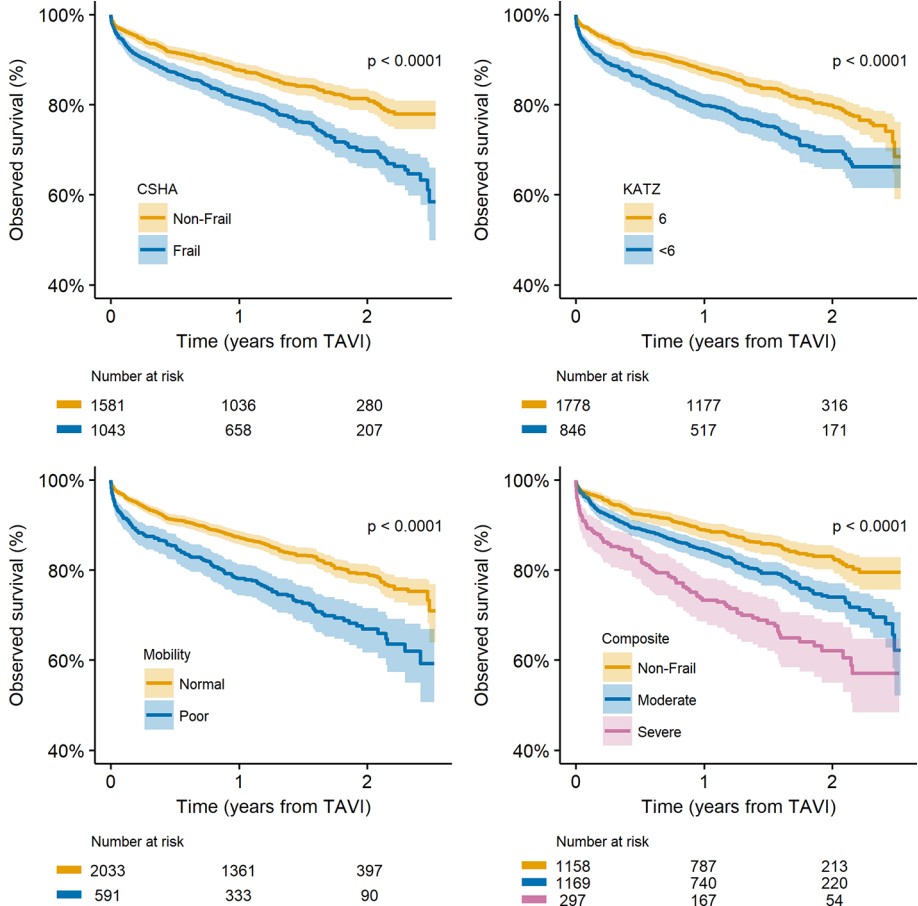

**Figure 2** Kaplan-Meier plots across CSHA-estimated frailty (top-left), KATZ activities of daily living dependency (top-right), physician-estimated poor mobility (bottom-left) and the composite frailty score (bottom-right). Definitions are described in table 1. CSHA, Canadian Study of Health and Ageing; TAVI, transcatheter aortic valve implantation.

**Table 4** Time-dependent Cox proportional hazards models per frailty indicator

| Frailty-by-time interaction | Univariable HR (95% CI)* | Multivariable HR (95% CI)* |
|---|---|---|
| CSHA-estimated frail | | |
| 0–180 days | **1.56 (1.22 to 1.98)** | **1.29 (1.00 to 1.66)** |
| 180–365 days | **1.63 (1.12 to 2.38)** | 1.37 (0.94 to 2.01) |
| >365 days | **1.85 (1.32 to 2.60)** | **1.61 (1.14 to 2.29)** |
| KATZ <6 | | |
| 0–180 days | **1.71 (1.34 to 2.18)** | **1.55 (1.21 to 1.99)** |
| 180–365 days | **1.86 (1.28 to 2.71)** | **1.74 (1.19 to 2.55)** |
| >365 days | 1.25 (0.88 to 1.78) | 1.23 (0.86 to 1.75) |
| Physician-estimated poor mobility | | |
| 0–180 days | **1.74 (1.34 to 2.25)** | **1.50 (1.15 to 1.96)** |
| 180–365 days | **2.06 (1.39 to 3.06)** | **1.84 (1.24 to 2.75)** |
| >365 days | **1.50 (1.03 to 2.20)** | 1.36 (0.92 to 2.00) |

*Bold items indicate significant results.
CSHA, Canadian Study of Health and Ageing.

adjusted hazards were not significantly different for these measures. CSHA-estimated frailty had significantly increased hazards of mortality 0–180 days and >365 days post-TAVI, but hazards were not significantly different between these time windows. Quantitatively, similar results were found across the composite score frailty groups.

### Addition of frailty in TAVI CPMs

Table 5 gives the discrimination of each TAVI CPM for predicting 30-day mortality both before and after the addition of each frailty measure. The LRT indicated a significant improvement in model fit by adding CSHA-estimated frailty into the FRANCE-2 model (p=0.037), the OBSERVANT model (p=0.020) and the ACC model (p=0.048). Adding KATZ <6 into the TAVI models significantly improved the AUC of both the FRANCE-2 CPM (p=0.047) and the OBSERVANT CPM (p=0.007), with the LRT indicating a significant improvement in model fit for all three TAVI CPMs (p<0.001). Equally, physician-estimated poor mobility significantly improved the fit of all three models (LRT p<0.001) and significantly increased the AUC of the OBSERVANT (p=0.006) and the ACC (p=0.030) CPMs by 7% and 5%, respectively. A forward stepwise selection of all three measures resulted in KATZ <6 and physician-estimated poor mobility being added into each of the existing TAVI CPMs (table 5), that is, CSHA-estimated frailty did not significantly improve the predictive performance of the existing TAVI models after inclusion of KATZ <6 and physician-estimated poor mobility.

Finally, the absolute-agreement ICC between the original versions of each TAVI CPM was 0.39 (95% CI 0.27 to 0.49), which increased to 0.65 (95% CI 0.63 to 0.67), 0.78 (95% CI 0.76 to 0.79) and 0.81 (95% CI 0.79 to 0.82) on adding CSHA-estimated frailty, KATZ <6 or physician-estimated poor mobility, respectively. Thus, the patient-level

**Table 5** Discrimination of each TAVI CPM at predicting 30-day mortality before and after the addition of each frailty measure

| Model | AUC (95% CI) | P values* |
|---|---|---|
| FRANCE-2 | | |
| Original | 0.62 (0.57 to 0.68) | N/A |
| Updated with CSHA only | 0.64 (0.58 to 0.69) | 0.412 |
| Updated with KATZ only | 0.67 (0.61 to 0.72) | 0.047 |
| Updated with poor mobility only | 0.67 (0.62 to 0.72) | 0.058 |
| Updated with stepwise selection† | 0.68 (0.63 to 0.73) | 0.025 |
| Observant | | |
| Original | 0.56 (0.50 to 0.62) | N/A |
| Updated with CSHA only | 0.59 (0.53 to 0.65) | 0.129 |
| Updated with KATZ only | 0.62 (0.56 to 0.68) | 0.007 |
| Updated with poor mobility only | 0.63 (0.57 to 0.69) | 0.006 |
| Updated with stepwise selection† | 0.64 (0.58 to 0.71) | 0.002 |
| ACC | | |
| Original | 0.63 (0.58 to 0.68) | N/A |
| Updated with CSHA only | 0.64 (0.59 to 0.69) | 0.605 |
| Updated with KATZ only | 0.67 (0.61 to 0.72) | 0.069 |
| Updated with poor mobility only | 0.68 (0.63 to 0.73) | 0.030 |
| Updated with stepwise selection† | 0.68 (0.63 to 0.74) | 0.036 |

*The DeLong p value compares the AUC of predicting 30-day mortality using the original TAVI CPM with that using each corresponding updated version.
†Forward selection across the three frailty measures resulted in KATZ <6 and physician-estimated poor mobility being added into each existing TAVI CPM.
ACC, American College of Cardiology; AUC, area under the curve; CPM, clinical prediction model; CSHA, Canadian Study of Health and Ageing; N/a, not applicable; TAVI, transcatheter aortic valve implantation.

agreement in risk prediction between the three TAVI CPMs was significantly improved after adding any of CSHA, KATZ or poor mobility into the models.

### DISCUSSION

The findings from this study can be summarised as follows: (1) KATZ <6 and physician-estimated poor mobility were independently associated with increased 30-day mortality, and all three considered frailty measures were associated with increased VARC-2 composite early safety; (2) patients with KATZ <6 and physician-estimated poor mortality had significantly worse survival up to 1-year post-TAVI; (3) the addition of KATZ and physician-estimated poor mobility into existing TAVI CPMs significantly improved the

discrimination and patient-level risk-stratification agreement of the models.

## Frailty and clinical outcomes following TAVI

Given that frailty is an indicator of general health status and vulnerability among elderly patients,[7] it is naturally under study in the TAVI field. The current study supports and expands the findings from previous studies that have shown associations between frailty and poor TAVI outcomes.[8–13 32–34] For instance, a previous analysis of a single UK centre showed that poor mobility strongly predicted survival following TAVI[10]; the current study supports this finding at the national level, in showing that the odds of 30-day mortality in those with physician-estimated poor mobility were 2.15 times higher compared with normal mobility. Similarly, a multicentre investigation of the Japanese registry found that a semiquantitative Clinical Frailty Scale was associated with increased 30-day and 1-year mortality.[12] Together, the current and previous results suggest that frailty/disability is an indicator of worse outcomes post-TAVI, despite the heterogeneity in frailty measures used across the existing evidence base. AHA/ACC guidelines currently indicate that KATZ, gait speed and grip strength should be used in the assessment of surgical risk,[35] with the emerging data suggesting that similar recommendations should be made for TAVI risk assessment and prediction.

However, implementing this emerging evidence may prove difficult without consensus regarding the most clinically useful measure of frailty in TAVI patients. By comparing the predictive performance of seven different frailty scales, Afilalo *et al* recently recommended that the 'Essential Frailty Toolset', which includes multiple domains of frailty (motor skills, cognition and nutritional/physiological factors), be used in TAVI patients.[11] Data on such frailty domains were unavailable for the current study, with the three subjective measures collected within the UK TAVI registry mainly using measurements of ADL. Nonetheless, it is conceivable that assessment of ADL (eg, KATZ) could provide a simple mechanism of informing part of the risk assessment process for TAVI, until more objective frailty scales are routinely collected. For instance, a previous multicentre study by Alfredsson *et al* showed that an objective measure of gait speed was associated with 30-day mortality following TAVI,[32] but such measures are rarely recorded in national registries, which would currently limit their use to aid risk prediction/stratification in TAVI patients.

Moreover, outcomes after TAVI should consider both mortality and quality of life in the context of the elderly patients who predominantly undergo TAVI. Previous work has shown associations between frailty and quality of life[36] and a contemporary TAVI CPM that was derived to predict poor outcome (defined as mortality and/or reduced quality of life) included measures of frailty, functional status and cognitive decline.[37 38] Within the current study, indicators of quality of life were not available, which should be noted as a limitation of the study.

Consequently, further studies are needed to investigate the effect of frailty on endpoints such as quality of life or hospital readmission, which are increasingly used as a measure of futility in TAVI patients.

## Prognostic utility of frailty

Prognostic risk prediction for TAVI is an ongoing research area, with existing TAVI CPMs reporting only moderate performance when validated.[4–6 16] We found that adding indicators of dependency in ADL (KATZ <6) or physician-estimated poor mobility significantly improved the predictive performance and patient-level risk-stratification agreement of the TAVI models. This supports previous work that has demonstrated improvements in predictive performance of the EuroSCORE and STS models through the addition of frailty measures.[11 12 39 40] Consequently, the heart team should consider frailty measures in addition to the multiple comorbidities that are reflected in existing risk scores, preferably by the inclusion of frailty directly into the risk prediction. We recommend that future iterations of existing and new TAVI risk models should include markers of frailty/disability. Arguably, one should regard frailty as a spectrum, rather than a binary phenomenon; however, the sample size of the current study restricted our ability to subclassify KATZ and CSHA into their individual components. Moreover, the marginal improvement in predictive performance demonstrated after the inclusion of frailty/disability measures into existing TAVI CPMs, suggests that these measures cannot overcome the need to discover novel risk factors in this patient cohort, or the need to predict endpoints other than mortality (eg, readmission or quality of life).

Additionally, the quantification of TAVI risk by using subjective measures of frailty could be problematic in situations of low inter-rater reliability. The current study demonstrated large spatial variability in the proportion of TAVI procedures conducted on 'frail' patients, but the subjectivity of CSHA and KATZ means that we were unable to separate genuine spatial differences from systematic variation in how different teams define/record these measures. This provides further indication for the need to refine the assessment of frailty in TAVI patients, since the current methods were derived for different purposes.

## Study limitations

The strength of this study is that it is a large, contemporary study, which has investigated the impact of frailty on outcomes and mortality prediction post-TAVI from a national perspective. However, several limitations need to be considered. First, only three subjective frailty measures were available, with objective frailty tests such as 6 min walk distance, grip strength or gait speed not recorded. Second, unmeasured confounders inherent in most observational studies potentially influence conclusions. Third, given that we could only analyse TAVI patients in 2013/2014, this retrospective study was underpowered to investigate endpoints with small event rates such as

bleeding or myocardial infarction. As the volume of frailty data increases, future studies will be able to investigate such outcomes. Fourth, the current study had a significant proportion of patients who were removed due to missing frailty measures and/or missing life status, which could potentially bias the results. However, the proportion of missing data in other variables was low and we implemented a multiple imputation procedure. Finally, given that this was a retrospective analysis of a national TAVI registry, we could not examine frailty in untreated patients with aortic stenosis, and neither could we compare the outcomes with patients treated through surgical aortic valve replacement. Thus, associations between frailty and the propensity for conservative treatment could not be explored, with corresponding potential for selection bias. Specifically, the analysis highlights potential measures to predict the expected outcomes after TAVI, but not the expected outcomes without TAVI.

## CONCLUSIONS

Frailty and disability, as estimated by KATZ, CSHA and poor mobility, was significantly associated with mortality and morbidity after TAVI. The predictive performance and patient-level risk-stratification agreement were significantly improved by updating existing TAVI CPMs to include measures of frailty/ADL dependency. Hence, physician-estimated frailty/disability measures could aid TAVI risk stratification, until scales that are more objective are routinely collected.

**Author affiliations**
[1]Farr Institute, Faculty of Biology, Medicine and Health, University of Manchester, Manchester Academic Health Science Centre, Manchester, UK
[2]Queen Elizabeth Hospital, Birmingham, UK
[3]The James Cook University Hospital, Middlesbrough, UK
[4]Keele Cardiovascular Research Group, Institute of Applied Clinical Science and Centre for Prognosis Research Group, Institute of Primary Care and Health Sciences, Keele University, Stoke-on-Trent, UK
[5]Academic Department of Cardiology, Royal Stoke Hospital, Stoke-on-Trent, UK
[6]Guy's and St Thomas' NHS Foundation Trust, London, UK
[7]Microsoft Research, Cambridge, UK

**Acknowledgements** We would like to acknowledge all participating centres for collecting the data and the National Institute for Cardiovascular Outcomes Research (NICOR) for providing the UK TAVI registry extract for this study.

**Contributors** GPM, MS and MAM made substantial contributions to the concept of the work in addition to performing the analysis. GPM drafted the initial version of the manuscript. GPM, MS, PFL, MAd, MG, JT, SRR, UTK, IB and MAM interpreted the results, revised the paper critically for important intellectual content and approved the final version of the paper. All authors agreed to be accountable for all aspects of the work.

**Funding** This research was funded by the Medical Research Council, through the Health e-Research Centre, University of Manchester (MR/K006665/1) and the North Staffordshire Heart Committee.

**Competing interests** None declared.

**Patient consent** Not required.

**Ethics approval** The National Institute for Cardiovascular Outcomes Research (NICOR) which includes the UK TAVI registry has support under section 251 of the National Health Service Act 2006 to use patient information for medical research without informed consent. Further ethical approval was not required under current National Health Service research governance arrangements, as all data analysed in the study were pseudonymised and contained no patient identifiable information.

**Provenance and peer review** Not commissioned; externally peer reviewed.

**Data sharing statement** The data that this research was conducted on are available from the National Institute for Cardiovascular Outcomes Research (NICOR) after undertaking an application process.

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
