## [Reviewer comments · BMJ Open]

ARTICLE DETAILS

TITLE (PROVISIONAL)	Do frailty measures improve prediction of mortality and morbidity following transcatheter aortic valve implantation? An analysis of the UK TAVI registry
AUTHORS	Martin, Glen; Sperrin, Matthew; Ludman, Peter; deBelder, Mark; Gunning, Mark; Townend, Jonathan; Redwood, Simon R; Kadam, Umesh; Buchan, Iain; Mamas, Mamas

VERSION 1 – REVIEW

REVIEWER	Pawel Kleczynski Associate Professor, 2nd Department of Cardiology, Jagiellonian University Medical College, University Hospital, Krakow, Poland
REVIEW RETURNED	03-Mar-2018

GENERAL COMMENTS	The manuscript by Martin et al aimed to assess the association of frailty evaluated by Katz Index, CSHA and poor mobility with TAVI outcomes and the prognostic utility of adding frailty into existing clinical prediction models like France2-model, Observant-model and ACC-model based on 2624 pts enrolled in UK TAVI registry. The manuscript is well written and tackles a hot topic of frailty in TAVI pts, especially when consecutive pts are enrolled, presenting a real world data and not 'sophisticated' data of RCTs. Authors concluded that frailty and disability was significantly associated with mortality and morbidity after TAVI. The predictive performance and patient-level risk-stratification agreement was significantly improved by updating existing TAVI CPMs to include measures of frailty Comments: 1. Introduction part is rather too long. Try to focus on most important aspects so that the readers can get a straight point of the aim of the study.2. In methods section, frailty grouping definitions could be placed in supplementary material (Table 1), because, again, the methods section seems to be too long.3. Try not to repeat data in the tables again in the text.4. Try consider putting Venn diagram and figure with proportion of frail/non frail pats in the supplementary material.5. In the main manuscript I would leave the KM figure and Tables 2-7.6. Discussion section does not have its flow, it's too long, try to update and shorten a bit.7. Authors highlighted appropriately study limitations. However, in my opinion, the subjectivity of presented frailty scales should also be mentioned.8. Authors selected a wide spectrum of frailty references but I'd like to see more recent ones, especially from 2017 and 2018 (PMID: 29422352, 28463403, 28302751, 28267475 etc..).
--

REVIEWER	A. Schoenenberger Department of Geriatrics, Switzerland
REVIEW RETURNED	05-Mar-2018

GENERAL COMMENTS	The authors investigated whether frailty measures were associated with worse outcomes following TAVI and whether these measure have the potential to improve existing clinical prediction models using data from a registry in the UK. I have the following comments:  1. It is a little bit presumptuous to say that there are only preliminary studies having investigated frailty for better prediction in TAVI patients and to present data from a registry with its limitations and using frailty measures, which have own limitations. Furthermore, the literature overview in the Introduction and the discussion of the current results in view of the existing knowledge is insufficient. The authors should do an effort to cite the relevant papers in the field of frailty and TAVI, what they currently don't. Besides, this is not the first paper investigating the improvement in risk prediction by combining frailty with conventional risk scores in TAVI patients (e.g., Schoenenberger AW, JACC Cardiovasc Interv. 2018 Feb 26;11(4):395-403). 2. A major limitation of the paper is the fact that all frailty measures are not real frailty measures. None of the used frailty measures is based on performance testing. The CSHA is a 7-point scale based on observer-rating. The Katz scale is a self-report scale, and measuring independence in activities of daily living rather than frailty. Poor mobility was also determined by observer-rating. In addition, mobility is only one of several components, when measuring frailty. The imprecision in the measurement probably explains why the Vann diagram (Figure 1) does not show much overlap between the 3 measures within the study population. The authors do mention this limitation, however, in my opinion, this is a severe limitation and should be mentioned appropriately. 3. The data origin from a registry. A typical problem inherent in registries are missing data. The authors declare the number of patients they had to exclude because of missing follow-up ($\approx 4\%$) and because any frailty measure was missing ($\approx 10\%$). In the remaining study population, they used an imputation method. The authors should state, how many missing data there were among other variables, in particular among the frailty measures. In addition, I think this is another limitation of this study, which is not adequately mentioned within the limitations section. 4. Cut-offs of frailty measures: were these cut-offs sample-driven, determined a priori or were previously published cut-offs used? 5. Results: might be somewhat more concise and to the point. Too many tables (e.g., 3 tables for baseline characteristics by the 3 "frailty" measures).
--

VERSION 1 – AUTHOR RESPONSE

Reviewer 1:

The manuscript by Martin et al aimed to assess the association of frailty evaluated by Katz Index, CSHA and poor mobility with TAVI outcomes and the prognostic utility of adding frailty into existing clinical prediction models like France2-model, Observant-model and ACC-model based on 2624 pts enrolled in UK TAVI registry. The manuscript is well written and tackles a hot topic of frailty in

TAVI pts, especially when consecutive pts are enrolled, presenting a real world data and not 'sophisticated' data of RCTs. Authors concluded that frailty and disability was significantly associated with mortality and morbidity after TAVI. The predictive performance and patient-level risk-stratification agreement was significantly improved by updating existing TAVI CPMs to include measures of frailty

1. Introduction part is rather too long. Try to focus on most important aspects so that the readers can get a straight point of the aim of the study.

Author response: We thank the reviewer for highlighting this. In response, we have removed several redundant aspects of the introduction, with the aim of focussing on the issues relevant to the current study. The flow of the introduction is aiming to guide readers through to the fact that frailty is not considered in TAVI risk prediction, which leads onto our study aim. We hope that this shorter version of the introduction makes this clearer for readers. The changes are too numerous to list here, but can be seen within the tracked changes of the manuscript.

2. In methods section, frailty-grouping definitions could be placed in supplementary material (Table 1), because, again, the methods section seems to be too long.

Author response: While we do agree with the reviewer that, on reflection, the methods section was too long, we feel that Table 1 is required within the main manuscript to overview the definitions of each frailty measure – we foresee this being a useful reference table as readers go through the paper. Nevertheless, to reduce the length of the methods section we have removed or re-written sections (see manuscript tracked changes). Additionally, and in light of the results also being too long (as per comments by reviewer 2), we have removed one of the analyses that looks at the proportion of frail/non-frail per total centre volume (previously figure 2 and supplementary figure 3). Finally, we have moved (the previously labelled) Table 2, Table 3 and Table 4 into the supplementary material.

3. Try not to repeat data in the tables again in the text.

Author response: Thank you for highlighting this; we have aimed to remove such instances, where possible. For example, the odds ratios were removed from the text on page 11, since these are presented in Table 3.

4. Try considering putting Venn diagram and figure with proportion of frail/non frail pts in the supplementary material.

Author response: We did consider moving the Venn diagram (figure 1) into the supplementary material. However, given the significant reduction in the number of tables elsewhere, we feel it is now justified to keep the Venn diagram within the main manuscript. The authors believe that this figure helps illustrate the subjective nature of the three frailty measures (as per reviewer 2 comments). However, we have removed the figure showing the proportion of frail/non-frail across centre volume (previously figure 2). We hope that the removal of some figures/tables will reduce the density of the methods/results.

5. In the main manuscript, I would leave the KM figure and Tables 2-7.

Author response: The authors thank the reviewer for this very valuable comment. We have followed this suggestion, but have additionally removed some of the tables in line with the comments made by reviewer 2. Specifically, the three tables that originally compared the baseline

characteristics across the three frailty measures (originally tables 2, 3, and 4) have been moved into the supplementary material.

6. Discussion section does not have its flow, it's too long, try to update and shorten a bit.

Author response: The authors agree with the reviewer that the discussion section was originally too long. To aid structure, we have added more sub-headings within the discussion. Additionally, we have removed redundant sentences, where possible, to reduce the length. Moreover, a large proportion of text was devoted to discussing the findings from the centre volume and frailty proportions (originally figure 2); since this analysis has been removed, the corresponding text within the discussion has also been removed (page 16). Please see the tracked version of the manuscript to see specific changes.

7. Authors highlighted appropriately study limitations. However, in my opinion, the subjectivity of presented frailty scales should also be mentioned.

Author response: We thank the reviewer for this comment. We have now highlighted that the frailty measures were subjective in the limitations section of the discussion, and in the article summary section. The discussion now reads:

“Firstly, only three subjective frailty measures were available, with objective frailty tests such as 6-min walk distance, grip strength or gait speed not recorded.”

Similar wording was used in the article summary section:

“Only three subjective frailty measures were available, with objective frailty tests not recorded”

8. Authors selected a wide spectrum of frailty references but I'd like to see more recent ones, especially from 2017 and 2018 (PMID: 29422352, 28463403, 28302751, 28267475 etc..).

Author response: We thank the reviewer for highlighting these important references, which were originally omitted. Each of these has now been added into the paper to update our literature to evidence that is more contemporary (see reference numbers 12, 13, 34, 39 and 40).

Reviewer 2:

The authors investigated whether frailty measures were associated with worse outcomes following TAVI and whether these measure have the potential to improve existing clinical prediction models using data from a registry in the UK. I have the following comments:

1. It is a little bit presumptuous to say that there are only preliminary studies having investigated frailty for better prediction in TAVI patients and to present data from a registry with its limitations and using frailty measures, which have own limitations. Furthermore, the literature overview in the Introduction and the discussion of the current results in view of the existing knowledge is insufficient. The authors should do an effort to cite the relevant papers in the field of frailty and TAVI, what they currently don't. Besides, this is not the first paper investigating the improvement in risk prediction by combining frailty with conventional risk scores in TAVI patients (e.g., Schoenenberger AW, JACC Cardiovasc Interv. 2018 Feb 26;11(4):395-403).

Author response: We thank the reviewer for this comment. Firstly, we agree that the wording referring to existing/previous studies was too strong given the recently emerging evidence base of frailty-

TAVI analyses. In response to this comment, we have revised the wording within the abstract to remove the phrase “Preliminary studies..” and replaced this with “Previous studies...”. However, the current study is amongst the largest to date, which represents a key strength of our study.

Secondly, the references within the introduction and discussion have been revised according to the comments made by reviewer 1 (see reference numbers 12, 13, 34, 39 and 40). We hope that these also address the concerns raised by this reviewer regarding our reference list.

Finally, the authors thank the reviewer for providing the reference that has looked at adding frailty into the EuroSCORE and STS models (Schoenenberger AW, JACC Cardiovasc Interv. 2018 Feb 26;11(4):395-403). This reference was not published at the time of our initial submission, but we have now added this into the paper (see reference number 40). It should be noted that our paper supports the findings of the reviewer in their aforementioned publication, and the authors feel the current paper builds upon this evidence by examining the addition of frailty into existing TAVI-specific risk models, which (to the authors knowledge) has not previously been explored.

2. A major limitation of the paper is the fact that all frailty measures are not real frailty measures. None of the used frailty measures is based on performance testing. The CSHA is a 7-point scale based on observer-rating. The Katz scale is a self-report scale, and measuring independence in activities of daily living rather than frailty. Poor mobility was also determined by observer-rating. In addition, mobility is only one of several components, when measuring frailty. The imprecision in the measurement probably explains why the Venn diagram (Figure 1) does not show much overlap between the 3 measures within the study population. The authors do mention this limitation, however, in my opinion, this is a severe limitation and should be mentioned appropriately.

Author response: The authors completely agree with the reviewer that the frailty measures that are recorded within the UK TAVI registry are subjective in their assessment, and that this represents a limitation of our study. As such, throughout the paper, we refer to the measures in a way to emphasise this. Specifically, we systematically refer to the CSHA as “CSHA-estimated frailty”, the KATZ scale as “Activities in Daily Living dependency”, and poor mobility as “physician-estimated poor mobility”. Such wording aims to emphasise that these measures are both subjective and small components of a patient’s overall “frailty”. We have added the following text to the results section (page 10) to highlight the disagreement in the Venn diagram, which is potentially due to subjectivity in the frailty measures used:

“The disagreement in Figure 1 may relate to both the relative imprecision and subjectivity of each frailty measure and that the different measures assess disparate aspects of frailty.”

However, as noted by the reviewer, we have highlighted the subjective nature of the three measures in the limitations section. The subjectivity is also raised at several places within the discussion (e.g. page 14, 15, 16). Consequently, the authors are unsure what additional aspects the reviewer would like to be “mentioned appropriately”. Moreover, we disagree that the subjective nature of the frailty measures is a “severe limitation” since these are three metrics of frailty/disability that are being routinely collected in a national registry, and are amongst the most commonly used measures of frailty in the literature. Therefore, we propose that these measures are considered in TAVI risk prediction, until there is routine collection/recording of measures that are more objective. Using objective measures would indeed be preferable, but the lack of their routine collection currently restricts their use to aid risk prediction/stratification in TAVI patients. We state this viewpoint in our conclusion:

“Physician-estimated frailty measures could aid TAVI risk stratification, until more objective scales are routinely collected”.

3. The data origin from a registry. A typical problem inherent in registries are missing data. The authors declare the number of patients they had to exclude because of missing follow-up ($\approx 4\%$) and because any frailty measure was missing ($\approx 10\%$). In the remaining study population, they used an imputation method. The authors should state, how many missing data there were among other variables, in particular among the frailty measures. In addition, I think this is another limitation of this study, which is not adequately mentioned within the limitations section.

Author response: We thank the reviewer for highlighting this. While generally registry data are indeed subject to problems with missing data, the UK TAVI registry has a relatively low proportion of such cases. The precise proportions of missing data for each variable have now been added into Table 2. From this, we can see that the majority of variables had less than 1% of missing data. We have added a comment on the high proportions of missing follow-up and frailty measures to the limitations section, as follows (page 17):

“Fourthly, the current study had a significant proportion of patients who were removed due to missing frailty measures and/or missing life-status, which could potentially bias the results. However, the proportion of missing data in other variables was low and we implemented a multiple imputation procedure.”

4. Cut-offs of frailty measures: were these cut-offs sample-driven, determined a priori or were previously published cut-offs used?

Author response: As stated within the methods (page 6), the cut-offs used to dichotomise CHSA and KATZ was based primarily on the empirical median value, or from existing publications, where possible.

“Dichotomising CSHA and KATZ scores into two categories was based on the empirical median value (CSHA=“apparently vulnerable” and KATZ=6), and on the original publication of CSHA”

5. Results: might be somewhat more concise and to the point. Too many tables (e.g., 3 tables for baseline characteristics by the 3 “frailty” measures).

Author response: Thank you for this valuable comment. The number of tables reported has been significantly reduced, in line with similar comments made by reviewer 1. The tables that compare baseline characteristics across the three frailty measures have now been placed within the supplementary material (Supplementary Tables 2, 3 and 4). Additionally, we have removed the previous analysis that examined the proportion of “frail” patients per total centre volume (previously figure 2 and supplementary figure 3), with the aim of reducing the number of reported findings. The authors hope that this, in addition to several stylistic changes within the results section, makes this section clearer.

VERSION 2 – REVIEW

REVIEWER	Pawel Kleczynski Jagiellonian University Medical College, University Hospital, Krakow, Poland
----------	---

REVIEW RETURNED	13-Mar-2018
GENERAL COMMENTS	Thank you for implementation of several suggestions into manuscript. I have no further comments.
REVIEWER	Schoenenberger Andreas Department of Geriatrics, Bern, Switzerland
REVIEW RETURNED	16-Mar-2018
GENERAL COMMENTS	The authors did an effort to improve the manuscript and responded appropriately to my comments. I do not have any further comments.